# In Situ Study of the Painting "Hiroshima I" (1958) by Werner Tübke (1929–2004)

**Aleksandra A. Smolianskaia** [1,2,*], **Ivan I. Andreev** [1,2,*], **Sergey V. Sirro** [1,2], **Vladimir A. Aseev** [1], **Elena Y. Tereschenko** [1,3,4] and **Olga A. Smolyanskaya** [1]

1    Heritage Science Lab, ITMO University, 199034 Saint Petersburg, Russia; sirro2008@rambler.ru (S.V.S.);
     aseev@oi.ifmo.ru (V.A.A.); elenatereschenko@yandex.ru (E.Y.T.); smolyanskaya@itmo.ru (O.A.S.)
2    The State Russian Museum, 191086 Saint Petersburg, Russia
3    Shubnikov Institute of Crystallography of FSRC, "Crystallography and Photonics",
     Russian Academy of Sciences, 119333 Moscow, Russia
4    National Research Centre "Kurchatov Institute", 123182 Moscow, Russia
*    Correspondence: asmolyanskaya@yandex.ru (A.A.S.); iiandreev@itmo.ru (I.I.A.)

**Highlights:**

- A hidden portrait of Werner Tübke's father was revealed.
- The state of the painting, the texture of the author's brushstroke, the technique of applying the paint layer, the artist's work on the composition, and the types of pigments and binders used were identified.
- The unique artistic language of Werner Tübke at an early stage of the artist's work was examined.
- The grounds have been laid for the further comparison of Tübke's early paintings with his later works, as well as his contemporaries' and predecessors'.

**Abstract:** This article presents the results of technical studies of the oil painting by the artist of German origin Werner Tübke "Hiroshima I" (1958). The creative heritage of this author has not been studied enough and represents scattered data on the technology of painting and artistic techniques. The aim of this work was to determine the art materials and painting technology described in his diaries, using the example of his only painting represented in Russia: "Hiroshima I". For this purpose, an in situ approach was implemented using some simple museum instrumentations—UV-induced visible luminescence, infrared reflectography (IRR), radiography, portable X-ray fluorescence (XRF), Fourier transform Infrared spectroscopy (FT-IR), and polarizing microscopy using microprobes. As a result, the pigment composition of the painting layers could be determined, the painting technology refined, and a previously unknown hidden portrait of Werner Tübke's father revealed.

**Keywords:** Werner Tübke; GDR art; Hiroshima; hidden image; FT-IR; XRF; diagnostic analysis; pigment identification

## 1. Introduction

This article focuses on a technical study of the creative legacy of Werner Tübke (1929–2004), one of the leading official artists of the German Democratic Republic (GDR) and, along with Wolfgang Mattheuer, Bernhard Heising, as well as Willy Sitte, the most outstanding representative of the Leipzig school of painting [1–3]. One of the many diary entries of Werner Tübke gives a faithful representation of this master: "I paint in stages. In the beginning, I make a false color in tempera mixed with chalk, then I add English colors: black, red, and white. I do not go too deep into perspective. In the second layer, I intensify the color with pearl glaze which I apply on the top of the darker shades. The white mixture is made up of tempera white and white oil. I then apply an intermediate coat of varnish from dammar, turpentine, and Venetian turpentine at a ratio of 60:30:10. It is only the third

coat that I start painting rationally in the wet-to-wet technique in one session until the colors are completely dry, using dammar, poppy oil, and turpentine in a 2:1:1 ratio" [4].

Tübke's adherence to the classical tradition of easel painting and his universal mastery of painting techniques is one of the unique features of his oeuvre. As an artist with higher not only artistic but also pedagogical education, Tübke brilliantly mastered the techniques of the old masters, disregarding the methods of pre- and post-war modernism [5].

Let us consider the famous "Peasant War Panorama" in Bad Frankenhausen (1976–1988), the monumental painting "The Working Class and the Intellectuals" at the University of Leipzig (1973); the triptych "Man is the Measure of All Things" for the Palace of the Republic in Berlin (1975) or his altarpiece in the church in Clausthal-Zellerfeld (1997) [6–9]. All of these pieces stand out for their intricate link to the traditions of fifteenth and twentieth-century painting, not only on the level of iconography and style but also in terms of technology. The artist himself described the process of creating each work in detail; he selected the base of the painting, and the type of primer, created the preliminary drawing, and selected the composition of binders, varnishes, and pigments. In this way, he was guided by the techniques of the old masters, among whom he was particularly influenced by Brueghel and the representatives of naturalistic realism of the late Middle Ages, the art of the fifteenth and sixteenth centuries in northern Europe, the German tradition of Dürer, Grünewald, the Cranach and Baldung, but also Italian Mannerism and its influences from Flanders to Spain, Goya and late Venetian painters including Magnasco, Tiepolo, and Longi. He never directly imitated the German Romantics, such as Friedrich and Runge; however, he constantly had the Nazarenes (especially Julius Schnorr von Carolsfeld, maybe indirectly Moritz von Schwind) and the early Düsseldorf school (Wilhelm Schadow) in mind [10].

All key experts on Tübke's art highlight the deep connection between Tübke's art and the traditions of the old masters. Günter Meissner, the main Tübke biographer, presents an extensive monograph entitled "Werner Tübke: Leben und Werke" [11], which describes Tübke's artistic education, his childhood and drawing studies in Schönebeck, his years at the art school in Magdeburg, at the University of Greifswald and at the University of Graphic Arts Leipzig, his career choice and early artistic quests, his teachers and the premises for his realistic style and historical thinking. Another serious scholar, Harald Behrendt's "Werner Tübkes Panoramabild in Bad Frankenhausen: Zwischen staatlichem Prestigeprojekt und künstlerischem Selbstauftrag" [12–14] has a depth and more attention to detail. The author does not only describe the twelve-year process of creating the monumental panorama in great detail but also gives an extensive commentary on the role of color and composition in the spirit of the old masters. Eduard Beaukamp dedicates the book "Werner Tübke. Arbeiterklasse und Intelligenz. Eine zeitgenössische Erprobung der Geschichte" [15] and numerous allusions to art from the past. Frank Zöllner also refers to them in the paper "History of the German Working Class Movement of 1961 and its Place within its Commissioned Art Works" [16]. The book "Zellerfelder Flügelaltar von Werner Tübke und seine Vorarbeite" [17] focuses on Tübke's later work, the Altar in Clausthal-Zellerfeld (1997), and includes a description of the artist's late technique. Finally, Rudolf Kober and Gerd Lindner devoted very informative and important research to the influence of Matthias Grunewald's painting on artists in the GDR and Tübke in particular [18]. Other publications could continue this list.

Nevertheless, despite the general understanding of the need to consider Tübke's heritage in the context of the great artistic tradition of European painting, at the level of analysis of the precise material used, a certain limitation of approaches is revealed. Most of the authors of publications on Tübke are interested in the iconography of his works and the context of their creation. Some of them move away from the art itself and focus on the ideological views of the state painter. For all the value of each statement, a rare author allows himself, after studying in detail this or that piece, to look at it "closely" and ask more specific questions. It seems to us that, in the case of such an adherent to the old tradition of easel painting as Tübke, a specific work of art and its individual characteristics should serve as the main source and material of research. Thus, the technical analysis of

his works could help us answer the question of what the originality of his art is, why it is so interesting to look at him today, and what the preconditions for his works are.

As the object of analysis, we took the only Tübke's painting in Russia, "Hiroshima I" (canvas, oil, NT-287, Figure 1), presented to the Russian Museum in 1995 by Peter Ludwig and exhibited in the Marble Palace [19]. This artwork represents a part of a series of three paintings on the same theme: "Hiroshima I", "Hiroshima II", and "Hiroshima III" (the two latter are held in Germany). Tübke's choice of subject matter is revealing: along with current events, Tübke's themes cover primarily historical subjects. In "Hiroshima I", for example, he reflects on the American atomic bombing of the Japanese city of the same name in 1945, namely, a destroyed cityscape, a chaotic mixture of bodies, people lying motionless or gesticulating, debris, and bicycle parts. Frank Zöllner describes the picture as follows: "From the mid-1950s onwards, Tübke then turned to politically more explicit subjects. In 1955 he began working on a painting to mark the tenth anniversary of the dropping of the first atomic bomb on Hiroshima, of which he eventually produced three versions. In these pictures, Tuübke invokes both his own apocalyptic fears and visions, and also the threat of nuclear war posed first and foremost—in East Germany's reading of the situation—by nuclear armament on the part of the West" [16] (p. 332).

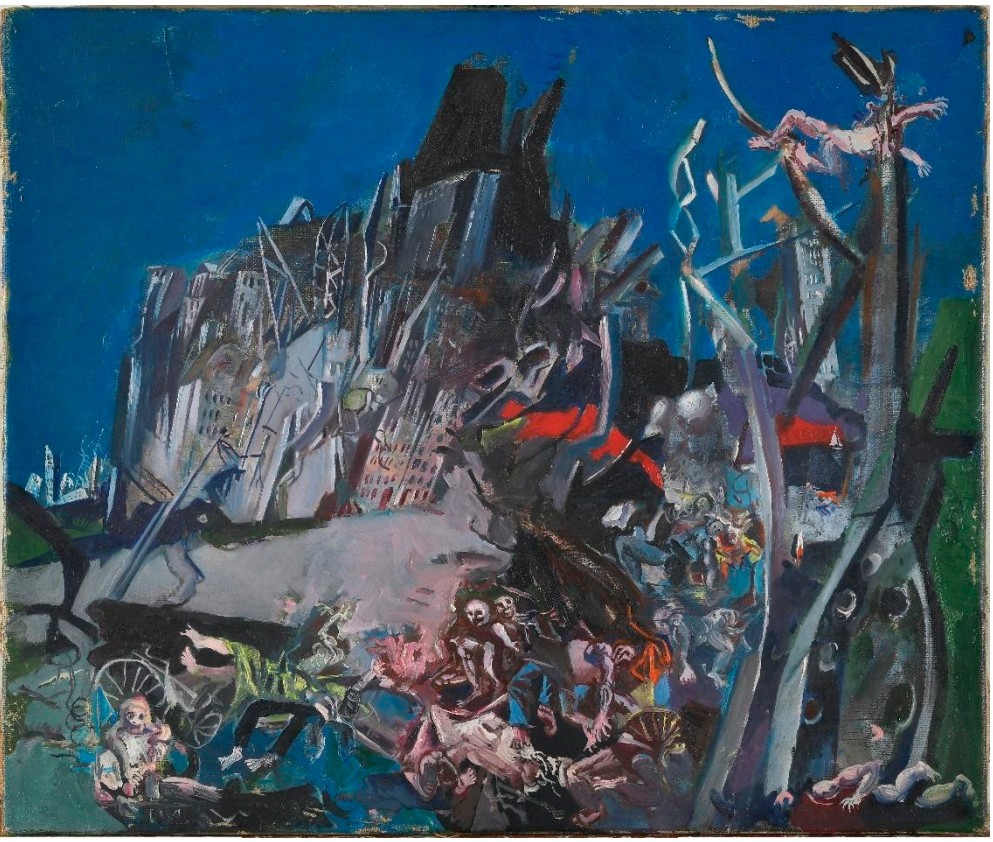

**Figure 1.** *Hiroshima*. Werner Tübke. Oil on canvas, 1958, 45.4 × 38.3 cm.

The painting was produced in 1958, at a time when the artist was making a decisive shift in the level of content as well as style. He departs from modernism towards figurative painting. This transition was most clearly reflected in the diptychs from the "Five Continents" cycle, which he painted in 1958 for the Astoria Hotel in Leipzig as a part of a government commission. It was the first time when he used the masters of the German Renaissance as his models so explicitly. The painting "Hiroshima I", on the other hand, represents a modernist experiment peculiar to his early work and is not typical for his mature period. Nevertheless, a laboratory study of this painting seems important because of the need to study Tübke's painting technique and the unique features of his paintings

(such as hidden images, the author's edits, and the pigment composition of the image) in order to understand and preserve its heritage for future generations.

The painting has never been studied before. It was not possible to take it outside the museum; therefore, it was examined with the equipment available at the Russian Museum's Technical and Technological Research Department. For the duration of the research, the painting was removed from its permanent exhibition in the Marble Palace and was brought to the laboratory in the Mikhailovsky Palace. In order to understand the special artistic features of Tübke, it was decided that an in situ study would be conducted [20–22] of his paintings using the following methods: optical microscopy, UV-induced visible luminescence, infrared reflectography, radiography, portable X-ray fluorescence (XRF), Fourier transform Infrared spectroscopy (FT-IR), and polarization microscopy.

## 2. Materials and Methods

### 2.1. Visual Analysis Methods

An X-ray integral image of the structure was obtained. An X-ray tube (Introvolt 100 VE, Promavtomatika) was placed under the object and directed to its front side. The X-ray tube's specifications were a voltage of 35 kV and a current of 4 mA. Consequently, the X-ray film (Agfa 100 NIF, Agfa) was over the object and directed to its reverse side. Pictures in visible light were performed using a Nikon D850 camera for macro photography with fixed lenses at 50 mm (f/10, 1/5 s, 200 ISO) and 200 mm (f/5, 1/5 s, 200 ISO).

The images, in their turn, were obtained by UV-induced visible luminescence using Master Alpha 16 UV 365 nm LED. The sources were located at a distance of one meter from the object and at an angle of 45 degrees. UV-induced visible luminescence was used to study the surface defects of the paint layer and the primer of the object. Nikon D800 with an infrared filter removed from a digital matrix, with a 50 mm lens (f/3.5, 1/2 s, 200 ISO), and Master Alpha 16 IR 950 nm LED sources were used to obtain infrared reflectography images [23]. The analysis of the painting surface and sampling was performed using a STEMI 2000 binocular stereomicroscope (Carl Zeiss Microscopy GmbH).

Artistic methods of visual analysis included a formal and contextual analysis of the painting on the basis of technical research, as well as work with documentary and iconographic sources (Werner Tübke's painting and diaries and the literature about the artist).

### 2.2. Qualitative Analysis Methods

Elemental analysis was performed using an energy-dispersive X-ray fluorescence spectrometer (TRACER 5, Bruker) with a thin graphene window, a rhodium anode (Rh), and a silicon drift detector cooled by Peltier elements. The results were obtained by analysis in the air and in the mode of 40 keV, 300 µA, software "Artax".

In order to analyze the composition of binders and pigments FT-IR spectrometer (TENSOR 37, Bruker) was used. The spectra were recorded in an ATR (MVP-Pro™, Harrick) mode in the range 4000–380 $cm^{-1}$ with a spectral resolution of 4 $cm^{-1}$. The preparation was performed by crushing the sample and placing it on the surface of a diamond crystal. Single-point ATR measurements were made by recording a total of 128 scans and averaging the resulting interferograms. All spectra were processed using an extended ATR correction (a single reflection, angle of incidence 45°, average refractive index 1.5), and, in some cases, non-informative regions of the spectra (2400–1850 $cm^{-1}$) were removed, and baselines were corrected. For the identification of substances, the obtained spectra were compared with the library data in the FT-IR spectrometer (ATR-LIB-COMPLETE).

To clarify the pigment composition of the colorful layers, in some cases, a polarizing microscope (PLM-2, LOMO Microsystems) was used, which is designed to study objects in transmitted, reflected, and polarized light.

## 3. Results

### 3.1. Preliminary Observations by Visual Analysis

A visual inspection of the painting allowed us to assess the state of preservation and technical features of the upper colorful layers [24]. It appeared that the underlying green layer of the painting could be viewed through the microscope (Figure 2a,b). The examination also revealed that this painting was partially scraped (erased) in some places up to the canvas itself, and there were neither signatures nor inscriptions found on the painting. Moreover, the canvas emerged to have a sparse binding and a thick layer of glue. Visual analysis showed no developed craquelure or lacquer film on the painting. In addition, it became possible to identify that the primer was made in a factory as it was preserved around the edges. A few scuffs on the edges of the canvas indicated the negligent storage of the painting before it became a part of the museum collection (Figure 2). Aggregates of lead and zinc salts of carboxylic acids were not observed on the surface of the painting layer.

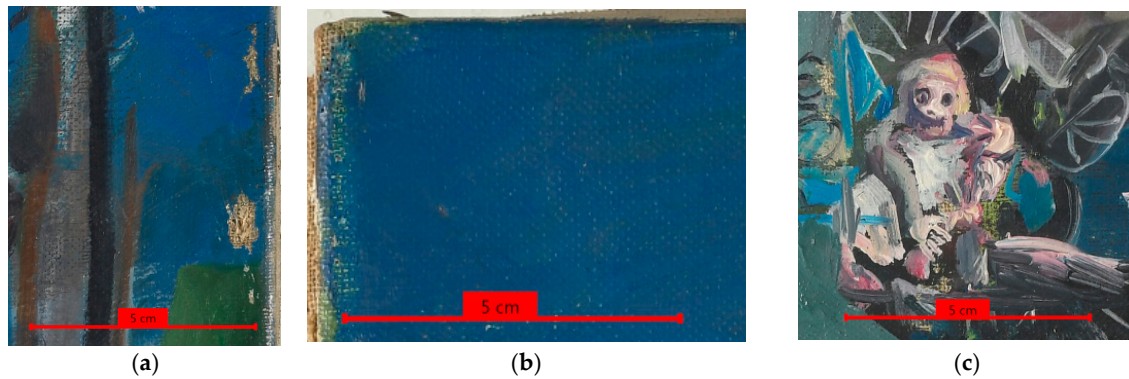

(**a**)　　　　　　　　　(**b**)　　　　　　　　　(**c**)

**Figure 2.** Microscope observation: scattering of colorful layers to the ground at the edges (**a**); Fragments of a green colorful layer under a new blue background (**b**); Scattering of colorful layers to the ground in the 3rd quarter of the painting (**c**).

### 3.2. Observations by UV Luminescence, IRR, Radiography

Photography in the UV-induced visible luminescence (Figure 3a) revealed the heterogeneous structure of the blue sky; moreover, it allowed luminescent pigments to be observed, which could be made up of organic as well as some inorganic pigments (for example, cadmium red). This assumption was later confirmed by the results of X-ray elemental analysis and Fourier transform infrared spectroscopy [24–26].

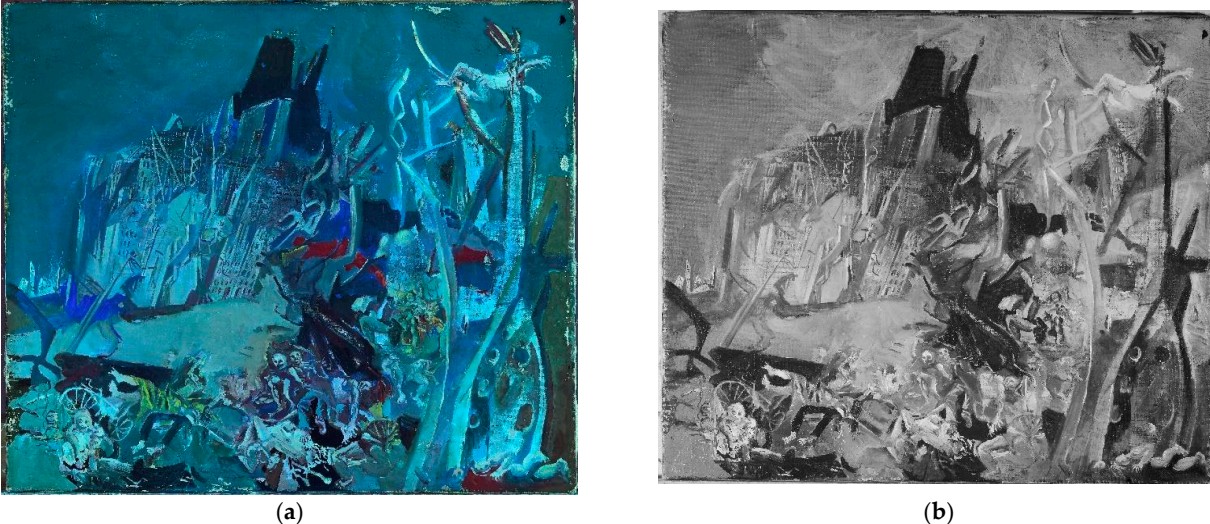

(**a**)　　　　　　　　　　　　　　　　(**b**)

**Figure 3.** UV-induced visible luminescence (**a**) and IRR (**b**).

Based on the results of infrared reflectography, no significant alterations were found. The underlying image could already be viewed in the photo (Figure 3b)—a portrait rotated 90 degrees clockwise.

To clarify the identity of the person depicted, a radiograph was obtained (Figure 4c) [27]. The presence of black and white areas on the radiograph, as well as gray transitions, indicated that the tube mode was selected correctly. To understand the location of the underlying image, similar areas were highlighted with three red rectangles (Figure 4).

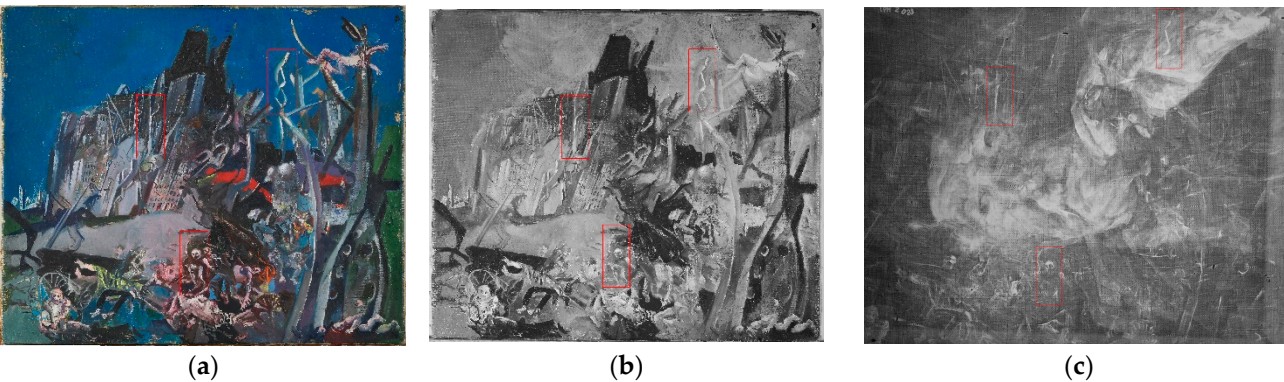

(**a**)                           (**b**)                           (**c**)

**Figure 4.** Markers on photo in visible light (**a**); IRR (**b**); Radiography (**c**).

The most recognizable person on the radiograph was Werner Tübke's father. Figure 5b,c shows two images of Werner Tübke's father for comparison with the artist's diary.

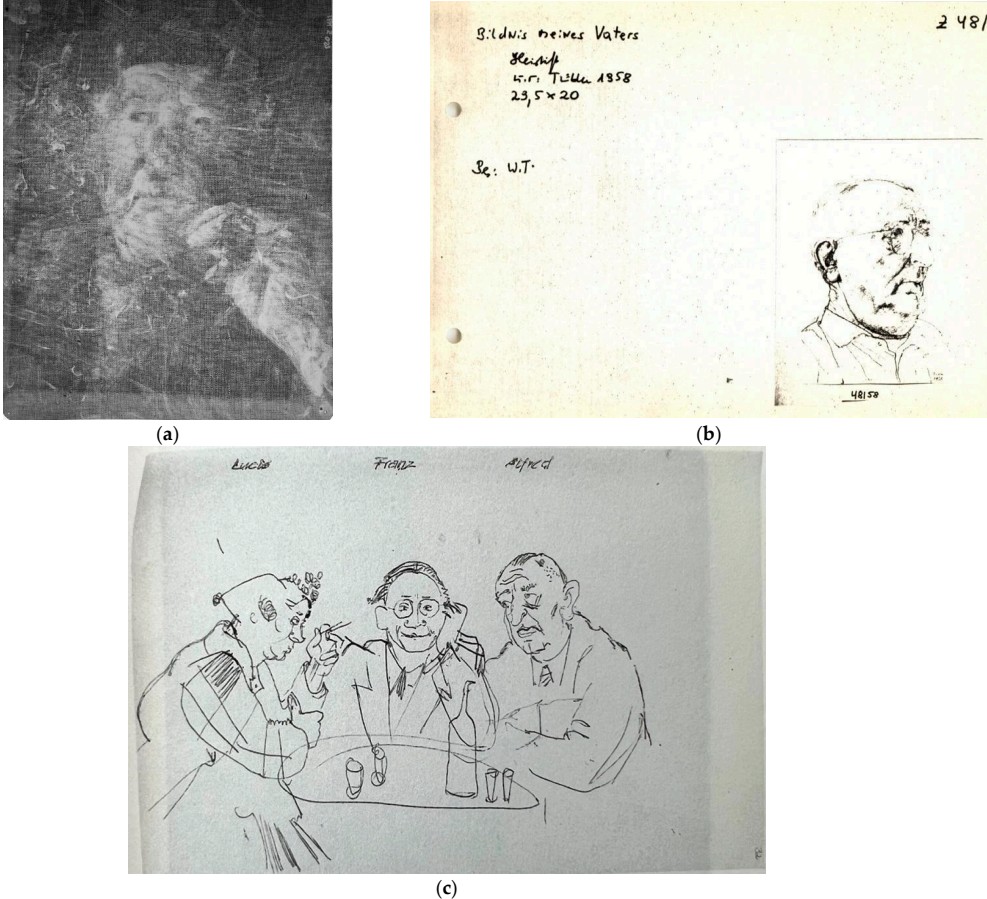

(**a**)                           (**b**)

(**c**)

**Figure 5.** Radiography of "Hiroshima I" rotated 90 degrees clockwise (**a**); "A portrait of my father. 1958" from a manuscript catalog of Werner Tübke, Panorama Museum Bad Frankenhausen (**b**); From

the left to the right in the Werner Tübke's picture: mother Lucie, Franz and father Alfred. 1952. Adapted with permission from Ref. [4] (p. 93). 2023, Michalski, A., Beaucamp, E. (**c**).

### 3.3. Elemental Analysis of Pictorial Layers (XRF)

The method of X-ray fluorescence was used to perform the non-destructive analysis of chemical elements [28]. This technique was performed at nine different points of the painting (Figure 6) to obtain meaningful data about the pigments and collect different chromatic shades that could define the artist's palette. X-ray fluorescence analysis (Table 1) revealed the presence of elements such as lead (Pb), zinc (Zn), iron (Fe), cadmium (Cd), selenium (Se), chromium (Cr), calcium (Ca), and barium (Ba). This canvas was coated with a primer containing lead, zinc, and calcium (Figure 6a). It may be lead white ($2PbCO_3 \cdot Pb(OH)$)—a pigment that has been widely used since ancient times both for making primers and for mixing different layers of painting [29]—as well as calcium, a typical filler for primers and paints, in the form of calcium carbonate ($CaCO_3$) or gypsum ($CaSO_4 \cdot (H_2O)_2$). The white was made with a mixed pigment containing zinc and lead [30,31] (Figure 7b); this was concluded based on the comparison of zinc and lead intensities with a primer.

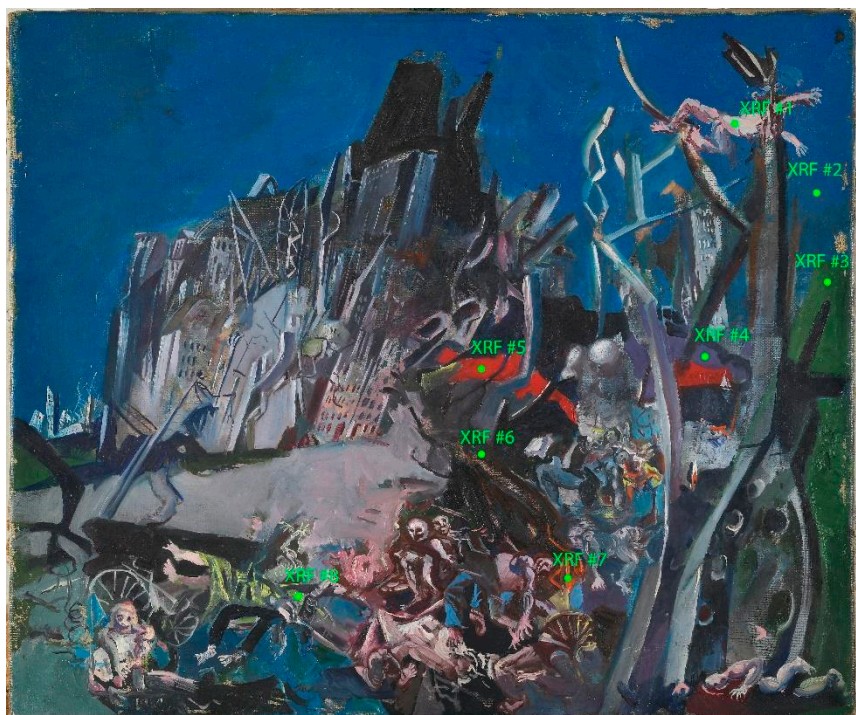

**Figure 6.** Scheme of the X-ray fluorescence measuring points (1–8) in the painting, and spot#9—white primer.

Chromatic pigments are represented by a wide palette, including organic pigments that could not be identified using this method. The inorganic red pigment was cadmium red (CdS x CdSe) (Figure 7c), a widely used pigment since the early 20th century, which was used as a substitute for the classic red pigment cinnabar (HgS) [31]. Moreover, mercury Hg was detected in #5, despite the presence of cadmium and selenium. Probably, the artist preferred to work with two inorganic red pigments. Some of the red pigments remained unidentified according to the results of X-ray fluorescence since no characteristic cadmium or mercury peaks were detected. To determine the above-mentioned pigments, the method of Fourier transform infrared spectroscopy was implemented.

**Table 1.** Qualitative analysis of chemical elements corresponding to nine investigated spots.

| Spot# | Colour | Main Elements |
|:---:|:---:|:---:|
| 1 | White | Pb, Zn |
| 2 | Blue | Pb, Zn, Fe |
| 3 | Green | Pb, Zn, Cr, Fe |
| 4 | Violet | Pb, Zn, Fe |
| 5 | Red | Pb, Zn, Cd, Se, Ba, Hg |
| 6 | Brown | Fe |
| 7 | Scarlet | Pb, Zn |
| 8 | Yellow | Pb, Zn, Cd, Ba |
| 9 | Primer | Ca, Zn, Pb |

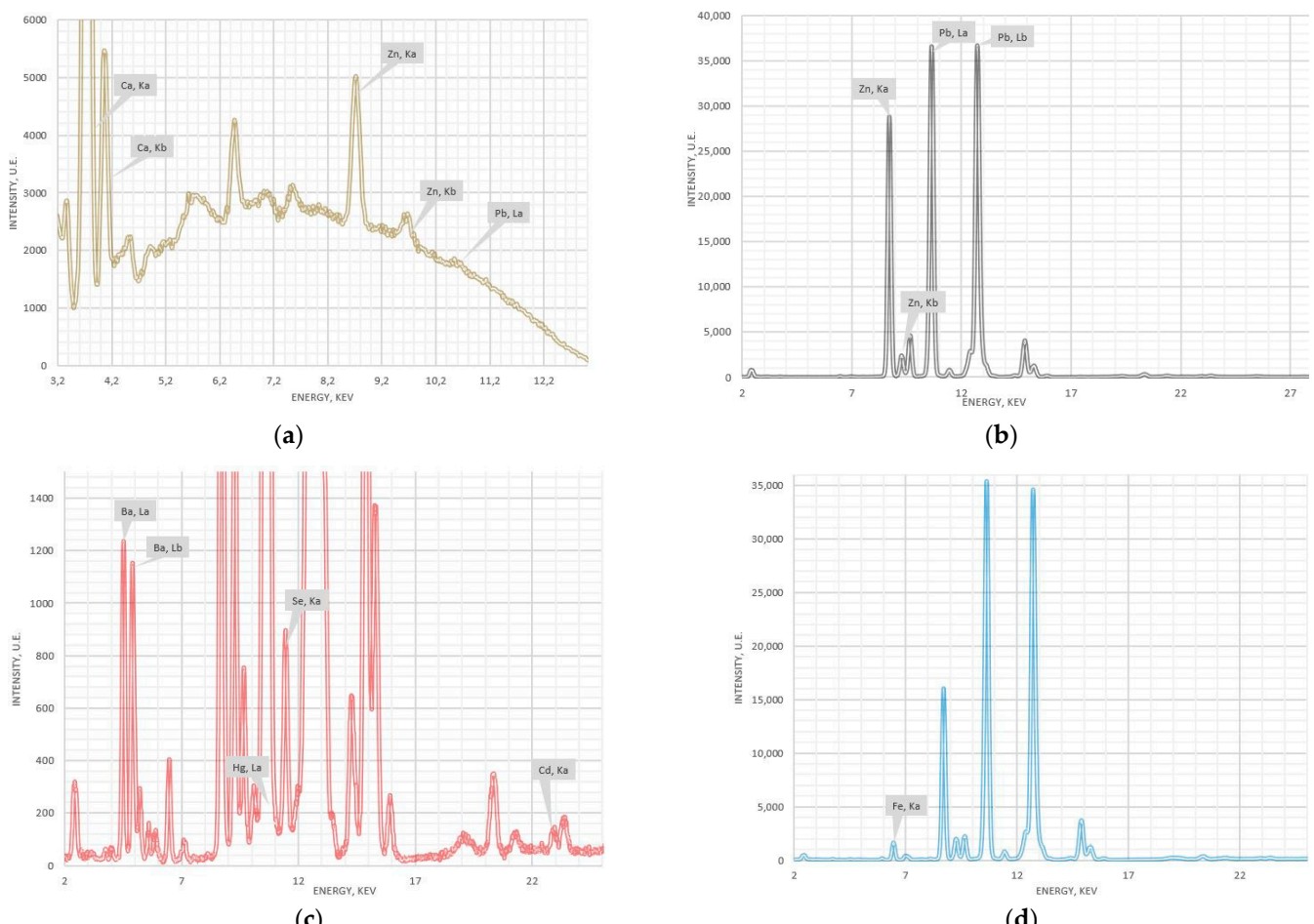

**Figure 7.** XRF spectra: spot#9, primer (**a**); spot#1, white (**b**); spot#5, red (**c**); spot#2, blue (**d**). The main peaks are marked.

The blue pigment was also unidentified, which suggested the use of either organic blue pigments, blue ultramarine ($Na_6Ca_2(AlSiO_4)_6(SO_4,S,CO_3)_2$), or Prussian blue ($Fe^{3+}_4[Fe^{2+}(CN)_6]_3$) (Figure 7d). The artist used cadmium-containing yellow as a yellow pigment [30]. The green pigment, based on spectral data, could be identified as a chromium-containing pigment. It could be chromium oxide ($Cr_2O_3$), chromium oxide hydrate ($Cr_2O_3\cdot H_2O$), volkonskoite ($CaO_3$ ($Cr^{3+}$, $Mg^{2+}$, $Fe^{2+})_2$ (Si, Al)$_4$ $O_{10}$ (OH)$_2\cdot4H_2O$) [32] or a mixture of blue and yellow chromate. The purple pigment may be purple mars ($Fe_2O_3$) due to the presence of iron in the spectrum. The

identification of this pigment was also performed using FTIR and polarization microscopy. Barium is widely distributed in the composition of an inert filler—barium sulfate ($BaSO_4$)—which is especially prevalent in cadmium paints [31].

Since the painting has underlying layers of unknown thickness, these results should be verified using additional methods. The current study used no methods to estimate the depth of the elements based on the absorption intensity of the K, L, and M series because the studied spot of the device was 1 cm in diameter [33]. The use of quantitative estimates of the spectral data based on X-ray fluorescence, in this case, was difficult due to the limitations of portable equipment [34].

### 3.4. FTIR-ATR Spectroscopy and Polarizing Microscopy

Using only XRF provided an incomplete understanding of the artist's color palette due to the physical limitations of the method. In this case, it seemed logical to use another spectral method—Fourier transform infrared spectroscopy (Figure 8). The absence of characteristic peaks for inorganic pigments on the XRF spectrum almost always refers to the use of molecular analysis methods. The infrared spectra of organic pigments from the painting and from the database are shown in Figure 9a,b respectively.

Alizarin Crimson main peaks: the stretching vibrations of the carbonyl group were between 1630 and 1670 $cm^{-1}$; aromatic $\nu(C-C)$ stretching vibrations occurred at ~1590 $cm^{-1}$; several bands were between 1420 and 1500 $cm^{-1}$, which could be assigned to combinations of vibrations of either $\nu(C-C)$, $\nu(C-O-H)$, $\nu(C-O)$, or $\delta(C-H)$ groups; stretching vibration of the $\nu(C-C)$ group occurred at ~1330 $cm^{-1}$ [35].

PY3: 1672s, 1614 w, 1593 ms, 1585 ms, 1566 ms, 1537s, 1504s, 1478 ms, 1442 ms, 1401 ms, 1358 ms, 1336s, the 1280s, 1260 ms, 1140 w, 962 w, 920 w, 892 w, 887 m, 812 m, 751 ms, 708 w, 675 w, 536 w, 483 w; $BaSO_4$: 1170 br, 1116 br, 1074 br, 982 ms, 634 ms, 610 s [35–37].

PR3: 1620 ms, 1562 ms, 1524 w, 1500 ms, 1448 ms, 1402 w, 1343 w, 1335 w, 1322 w, 1302 w, 1256 w, 1132 br, 987 w, 971 w, 925 w, 872 w, 850 w [36–38].

The presence of the oil component was confirmed by the intense absorption bands of asymmetric $\nu_{as}(-CH_2-)$ and symmetric $\nu_s(-CH_2-)$ stretching vibrations of the methylene group at 2919 $cm^{-1}$ and 2850 $cm^{-1}$, stretching vibrations of the carbonyl group $\nu(C=O)$ at 1736 $cm^{-1}$, and stretching vibrations of the ester bond $\nu(C-O)$ at 1160 $cm^{-1}$ [39]. An oil binder was present on all spectra of the paint layers.

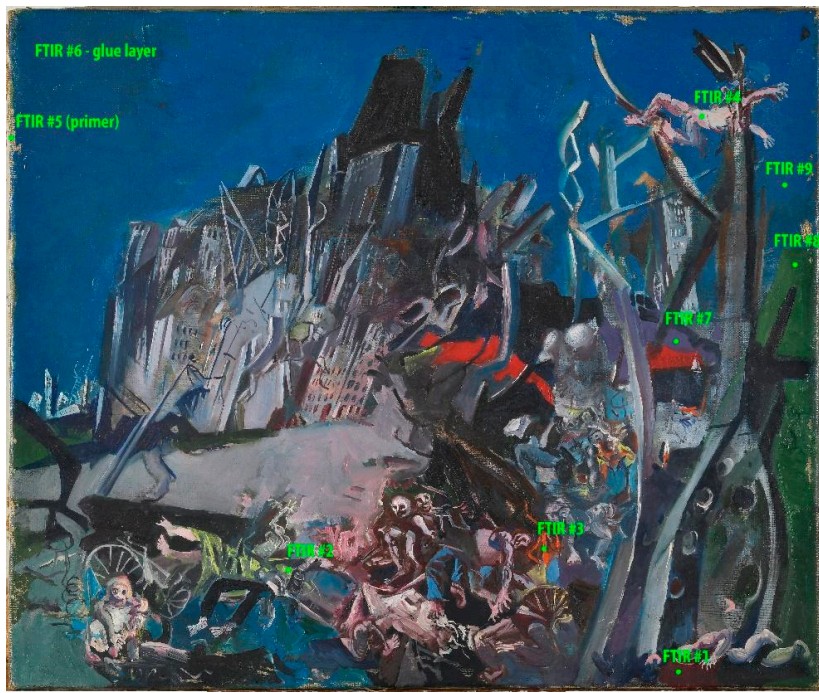

**Figure 8.** Scheme of the FTIR measuring spots (1–9) in the painting.

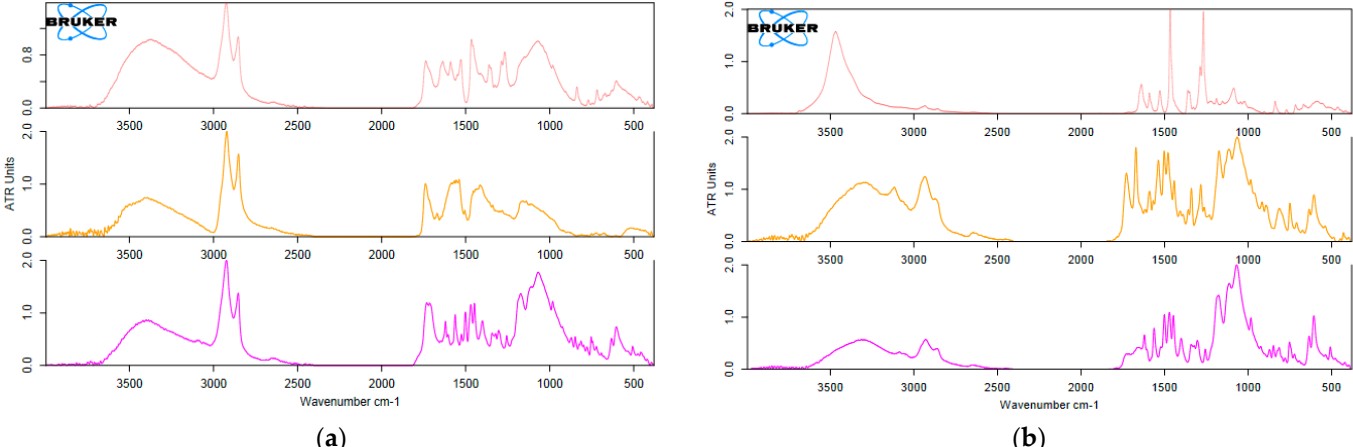

**Figure 9.** FTIR spectra: spot#1, red line (**a**); spot#2, orange line (**a**); spot#3, pink line (**a**); reference spectra of Alizarin Crimson—PR83, red line (**b**); PY3, orange line (**b**); PR3, pink line (**b**).

The white, according to the IR spectrum, corresponded to the composition of lead-zinc white ($2(PbCO_3)$ x $Pb(OH)_2$; $ZnO$). Currently, lead white is not produced on an industrial scale due to its high production toxicity. In the mid-infrared range, the IR spectrum (Figure 10a, blue line) showed the main absorption bands of the lead components, namely, the stretching vibrations of the hydroxyl group $\nu(OH)$ at 3537 cm$^{-1}$; the asymmetric $\nu_{as}(CO_3^{-2})$ and symmetric $\nu_s(CO_3^{-2})$ stretching vibrations of the carbonyl group at 1396 cm$^{-1}$ and 1045 cm$^{-1}$, respectively; and the out-of-plane $\delta(CO_3^{-2})$ and planar $\delta(CO_3^{-2})$ bending vibrations of the carbonyl group at 851 cm$^{-1}$ and 679 cm$^{-1}$, respectively [40]. It should also be noticed that zinc absorption bands presented in the low-frequency infrared region on the IR spectrum of the white paint (Figure 10a, blue line). They corresponded to a broad peak at 525 cm$^{-1}$, that is, the stretching vibrations of zinc oxide $\nu(Zn = O)$ [41]. The primer consisted of lead white, calcium carbonate, and zinc-containing pigments (according to XRF results). The binder was oil (Figure 10a, red line). The thick layer of gluing was gelatin (Figure 10a, pink line).

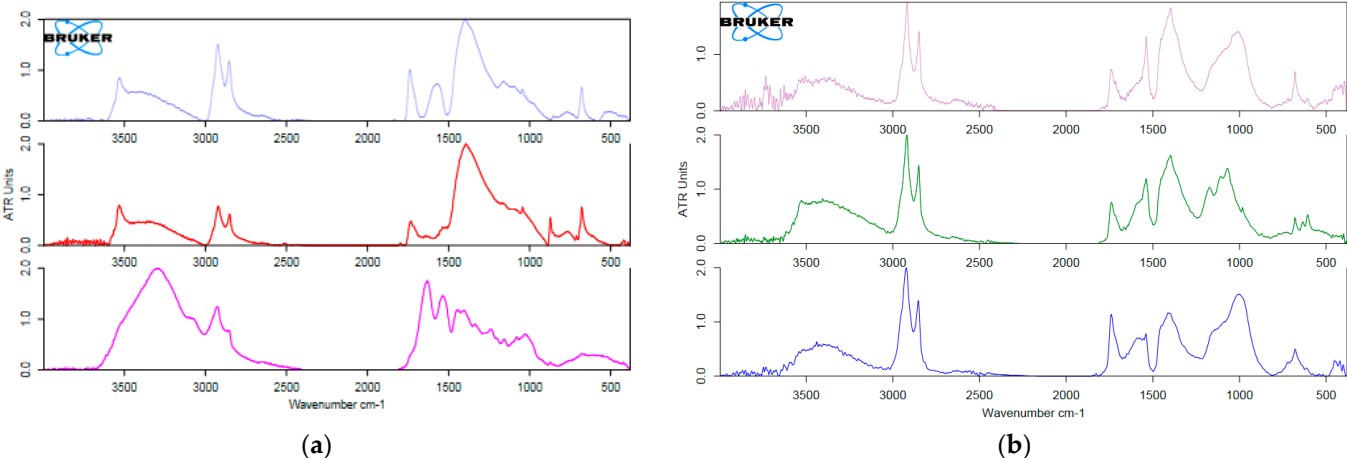

**Figure 10.** FTIR spectra: spot#4, blue line (**a**); spot#5, orange line (**a**); spot#6, pink line (**a**); spot#7, violet line (**b**); spot#8, green line (**b**); spot#9, blue line (**b**).

The blue pigment (spot#9), according to the research results, was ultramarine (Figure 10b, blue line). The purple pigment (spot#7) was quite difficult to determine by the infrared spectrum (Figure 10b, violet line). Based on the fact that its spectrum was almost identical to that of the blue pigment (ultramarine), we could assume the use of purple ultramarine or red mixed paint based on blue ultramarine. Thus, it became obvious that some additional research

was necessary to confirm this hypothesis. An analysis of the spectrum of the green pigment (spot#8) also revealed no characteristic absorption lines belonging to the green pigments (Figure 10b, green line). In order to identify it, polarization microscopy was used. For instance, we could suppose the presence of such a green pigment as volkonskoite, which was introduced at the beginning of the XIX century in Russia by Prince Volkonsky [33]. Figure 11 demonstrates the green pigment (a, b), as well as volkonskoite produced by the Leningrad Art paint factory in 1941 (c, d). The asymmetric stretching vibration $v_{as}(COO^-)$ of zinc and lead carboxylates was present on spectra #2,4,5,7,8,9 [42]. Moreover, the points of colorful pigments had a pronounced crystalline form of about 1538 cm$^{-1}$, in white and in the primer, which was amorphous at ~1585 cm$^{-1}$.

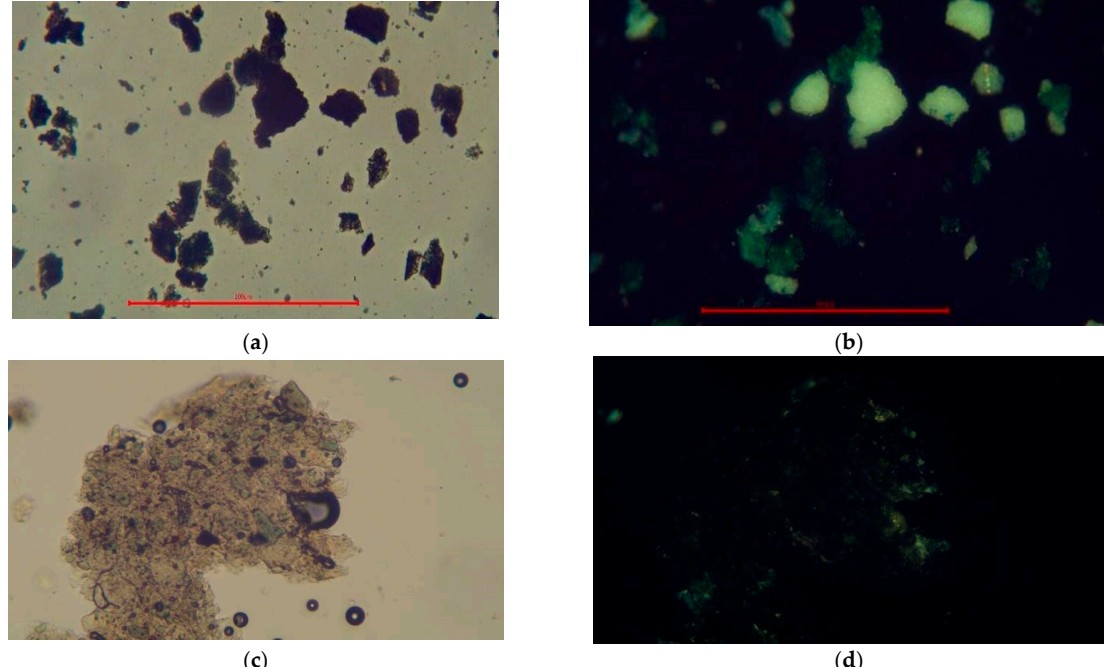

| (**a**) | (**b**) |
| (**c**) | (**d**) |

**Figure 11.** Photo of a green pigment in transmitted polarized light (**a**) and reflected polarized light with crossed polarizers (**b**); volkonskoite in transmitted polarized light (**c**) and reflected polarized light with crossed polarizers (**d**). Magnification 250×.

The metal salts of zinc and lead—carboxylates—were presented in all infrared spectra. It seems important to note that aluminum stearate was used as a surfactant after World War II and could interact with zinc white, increasing the concentration of the zinc stearate [42]. Table 2 presents the results of the studies on the pigment composition based on spectral data.

**Table 2.** General information about pigments.

| Colour | Identified Pigments | XRF Spot# | FTIR Spot# |
| --- | --- | --- | --- |
| White | Lead white, zinc white | 1 | 4 |
| Blue | Ultramarine blue | 2 | 9 |
| Green | Cr containing compound | 3 | 8 |
| Violet | Fe containing compound | 4 | 7 |
| Red | Cd, Se, Hg containing compound, PR81 | 5 | 1 |
| Brown | Fe containing compound | 6 | - |
| Scarlet | PR3 | 7 | 3 |
| Yellow | Cd containing compound, PY3 | 8 | 2 |
| Primer | Lead white, zinc white, calcium carbonate | 9 | 5 |

## 4. Discussion

The use of an integrated interdisciplinary approach allowed us to describe the object under study. The use of non–invasive visualization methods—UV, IRR, and radiography—provided us with an opportunity for art critics and curators to look at an art object from various viewpoints. The revealing of the pigment composition and binders could be useful in the dating of paintings and also simplifies the preparation for the restoration process.

The determination of the green pigment appeared to be difficult due to the absence of characteristic peaks of pigment absorption in the mid-infrared region. Therefore, we could only assume the use of volkonskoite based on the data from polarization microscopy and the presence of chromium in the XRF spectrum. Typical pigments for painting in the 20th century were also identified: lead-zinc whitewash, red cadmium selenide sulfide, yellow cadmium sulfide, blue ultramarine, red litol PR3, and yellow hansa PY3.

All the results of the study could be significant not only in terms of assessing the condition and storage conditions of the "Hiroshima I" painting but also from an art historian's viewpoint since it demonstrates modernist experiments in the artist's oeuvre. Thus, preliminary observations by visual analysis clearly reveal the expressive manner of this piece, namely, the textured strokes and the absence of subtle scumble. In addition, the painting was made in a single stroke; this is the way many expressionists painted earlier and these days. At the same time, ultraviolet photography showed that even in this case, Tübke made changes to the composition of the painting, replacing the insignificant details.

Observations by UV luminescence, IRR, and radiography were of particular interest in this research. These methods demonstrated the underlying layer of the painting, leading to the assumption that another painting was concealed under the Hiroshima image. Radiography confirmed this hypothesis by revealing a male portrait beneath the top layer that had been partially scraped (erased), in places, to the canvas. A comparison of the resulting image with the drawings by Tübke at the time confirmed the art historian's guess: the portrait of the artist's father, namely Alfred Tübke, on a green background is hidden under Hiroshima [43]. The identification with the artist's father is justified by a drawing found in the original handwritten catalog of Werner Tübke, which his widow gave to Gerd Lindner, director of the Panorama Painting Museum in the German town Bad Frankenhausen, many years ago as a copy. Mr Lindner kindly shared it with the authors of this paper. The physiognomic resemblance of the portraits is obvious. We know only one other image of the artist's father—a cartoon sketch depicting Alfred Tübke with his wife Lucy (the artist's mother) and another man [5] (p. 93). The similarity between the found drawings and the portrait on the bottom layer of "Hiroshima I" is so great that we can confidently conclude that we are looking at the same image of the artist's father. In contrast to the expressive painting of the upper layer, this portrait was executed in a more substantive, precise, academic manner, which the artist developed over the years.

An unobvious connection between "Hiroshima I" and Tübke's later works were also revealed by the elemental analysis of pictorial layers (XRF), Fourier transform infrared spectroscopy (FT-IR), and polarization microscopy. In particular, the green pigment volkonskoite and (presumably) Gutankara violet were found among the pigments of Tübke paintings. The exact elemental composition of these pigments has induced a number of questions. Curiously enough, the same pigments have aroused the interest and bewilderment of German colleagues studying Tübke's artistic legacy. In the autumn of 2022, one of the authors of this paper traveled to the artist's homeland in Germany, including the Werner Tübke Archive in Leipzig and the Panorama Museum in Bad Frankenshausen. Gerd Lindner, where the director of the Panorama Museum, explained that Tübke's only similar technical study in Germany was of the "Peasant War Panorama" (1976–1988). For the conservation and restoration of the monumental canvas, expertise was conducted in the 1990s, and some data were obtained, primarily concerning its paint layer. A German colleague raised questions about the same pigments: volkonskoite and violet ochre; they were unable to identify them precisely. The authors of this paper suggest that Tübke used these pigments to imitate the paintings of the old masters [44,45]. This theory is currently

being refined; we plan to report on it in our further research. We plan to compare our results with those of colleagues to understand the evolution of the artist's painting technique and his connection to his predecessors at the level of technology.

Thus, even in such a nontypical for Tübke's heritage example as "Hiroshima I", laboratory analysis revealed how meticulously he approached painting early in his career. On the one hand, the upper layer of his paintings in the expressive technique can be regarded more as a sketchbook in relation to his later works. The portrait of his father, on the other hand, anticipates his painting of the 70s and 80s (the artist's heyday period) since it is executed in an academic manner and corresponds to the national German tradition of figurative painting and easel painting. It demonstrates that already in the 1950s, at a time of changes in cultural policy in the GDR with its course toward socialist realism and the increasing ideological pressure on art, Tübke was looking for a language he could speak freely, remaining a politically engaged artist. In his early encounters with both pre-war modernism and the old masters, Tübke always aimed for a statement that was not linked to ideology or a specific artistic movement and yet was also invisibly linked to past traditions (the portrait of his father). The artist seems to have been driven by a desire to preserve the national style within the conditions of memory destruction after the Third Reich, the Second World War, and the emergence of the GDR. It was a search for an answer to a question of what being a German artist in post-war conditions meant.

Therefore, the study is an example of how museum laboratory methods can be applied to the art historical analysis of paintings, which is more complex and meaningful than art and cannot be exhausted with the data of ideological discourse. Tübke's legacy appears to be all too original (no such artists existed in the Third Reich or the Soviet Union), and his works reveal an artistic freedom that prevents him from being recognized as an opportunistic artist [46]. This fact seems important when assessing and preserving an important testimony of German cultural heritage and Werner Tübke's artistic legacy in honesty.

## 5. Conclusions

To sum up, this is the first research to use museum laboratory instruments to investigate Werner Tübke's painting "Hiroshima I". The researchers attempted to identify not only the state of the painting, the texture of the author's brushstroke, the technique of applying the paint layer, the artist's work on the composition, and the types of pigments but also to reveal a previously unknown painting by Werner Tübke (the portrait of his father). In addition, an academic study was discovered underneath the modernist painting, which characterized the young artist's search for his own unique artistic language at an early stage of his work (1958). It has been suggested that the artist already used volkonskoite and violet ochre in his paintings when he was imitating the Old Masters. For example, twenty years later, Tübke used the same pigments in the famous "Peasant War Panorama" in the German town Bad Frankenhausen, which has many references to the paintings of the past. Thus, the study of "Hiroshima I" allows a comparison of the pigment composition of Werner Tübke's early and late paintings. The conducted analysis gives grounds for the further investigation of this artist, little-studied but highly interesting in terms of painting technology, the traditions of German artistic heritage, and post-war art.

**Author Contributions:** Conceptualization, A.A.S.; Formal analysis, I.I.A. and V.A.A.; Supervision, S.V.S.; Project administration, E.Y.T. and O.A.S. All authors have read and agreed to the published version of the manuscript.

**Funding:** This research was performed with the support of a grant under the Decree of the Government of the Russian Federation No. 220 of 9 April 2010 (Agreement No. 075-15-2021-593 of 1 June 2021).

**Acknowledgments:** The authors would like to thank the management of the State Russian Museum, the staff of the Technical Department of the Russian Museum, the colleagues of the Contemporary Art Department of the Russian Museum and, above all, Marina Vladimirovna Seregina, senior researcher

of the Contemporary Art Department of the Russian Museum, and curator of the painting "Hiroshima I" in the Ludwig Museum's collection at the Russian Museum. We are also very grateful to Gerd Lindner, the director of the Bad Frankenhausen Panorama Museum for his professional cooperation and kindly provided materials.

**Conflicts of Interest:** The authors declare that they have no known competing financial interest or personal relationship that could have appeared to influence the work reported in this paper.

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
