# Peer review of "In Situ Study of the Painting “Hiroshima I” (1958) by Werner Tübke (1929–2004)"

_heritage, doi:10.3390/heritage6060255_

Round 1

Reviewer 1 Report

Dear Colleagues, your paper needs significant improvement.

The methodology of research is described by you in insufficient detail. Your readers should be able to replicate your experiments and evaluate the practical applicability of the methods you used based on the information in your article. Please indicate which research methods were quantitative, which were qualitative, and which were visual analysis methods.

For infrared reflectography, you used a Nikon D800 with an infrared filter removed from a digital matrix and Master Alpha 16 IR 950 nm LED sources. Here it is necessary to describe the shooting conditions in more detail, indicating the focal length and the exposure used. In this case, as a laboratory equipment for which a SWIR camera should be used, you used a converted camera that does not have the proper laboratory certification.

Elemental analysis was performed using a TRACER 5 energy dispersive X-ray fluorescence spectrometer (Bruker). Please indicate how you separated the signals associated with the paint layers and the ground of the painting you studied. Is such separation of signals possible or impossible when using this type of spectrometer?

FT-IR spectroscopy - please indicate the mass of the samples of the substance used for this type of analysis

Optical microscopy - in addition to a simple enumeration of the instruments used, it is necessary to indicate at least the degree of magnification used in the analysis of the surface of the picture and sampling from the paint layer.

Radiography - this research method is not described at all, even the brand of the device you used is not indicated.

Section 3.2. Observations on UV luminescence, IR radiation, radiography

Your reasoning in the context of evaluating the results of photography “about luminescent pigments, which in most cases are organic compounds” is methodically naive and incorrect - photography in the light of visible luminescence is not a method of chemical analysis and does not allow you to get the declared result.

The authors' reasoning about the portrait that exists under the colorful layers of the studied painting is based on the results of X-ray analysis, which is not included in the description of the research methods used.

Section 3.3. Elemental analysis of imaging layers (XRF)

The results of the determination of pigments by the XRF method declared by you are incorrect and do not correspond to the research method used. Pigment determinations can be obtained by XRD or Raman spectroscopy, which can take the form of a non-destructive method. Modern XRF analysis with the determination of the quantitative elemental composition and elemental mapping allows us to assume the presence of certain types of pigments. But the authors defined only chemical elements, and only in the form of the simplest qualitative, not quantitative analysis. Operating with the results of the XRF method, in its simplest manifestation, you are actually engaged in the intuitive reconstruction of pigments, replacing the result of instrumental studies with your ideas about pigments. For example, the formula of lead white given by you can be confirmed only by the results of phase analysis carried out by the XRD method, or by the results of Raman spectroscopy, but not by the results of XRF analysis in its simplest qualitative form.

The results of the study of the paint layer by optical microscopy are not confirmed by anything except for verbal declarative statements. Two shots of medium quality green pigment (Figure 8) are clearly not enough.

Section 4. Conclusions.

Some of the conclusions stated in the paper do not correspond to the results of the study.

The authors claim that they have also identified pigments typical of 20th-century painting: lead-zinc white, red cadmium sulfide selenide, cadmium sulfide yellow, ultramarine blue, lithol red PR3, Hansa yellow PU3. But you could not get such a result, given that you used XRF in its simplest form as the main method for identifying pigments.

In general, the work is more of an art history, rather than an interdisciplinary study. In the work, it is the aspect of art history that is most vivid and informative, and the instrumental research methods used by the authors are simple and uninformative, the methods of these studies are described in the work extremely briefly or not described at all (X-ray). Thus, the conclusions made by the authors on the composition of the pigments used by the master to create the work under study do not correspond to the results obtained and need to be corrected and improved.

Author Response

Manuscript ID: 2410352    
Title: Multidisciplinary study of the painting “Hiroshima I” (1958) by Werner Tübke (1929-2004)

First and foremost, we would like to thank the reviewers for their highly professional feedback which helped us improve the quality of the manuscript. Now, we would like to address the reviewers’ comments in the order of their appearance.

 --------Reviewer Comments--------

Reviewer 1:

The methodology of research is described by you in insufficient detail. Your readers should be able to replicate your experiments and evaluate the practical applicability of the methods you used based on the information in your article. Please indicate which research methods were quantitative, which were qualitative, and which were visual analysis methods.

Artistic methods of visual analysis have been added to the article. These include a formal and contextual analysis of the painting on the basis of technical and technological research, as well as work with documentary and iconographic sources (Werner Tübke's painting and diaries, literature about the artist).

For infrared reflectography, you used a Nikon D800 with an infrared filter removed from a digital matrix and Master Alpha 16 IR 950 nm LED sources. Here it is necessary to describe the shooting conditions in more detail, indicating the focal length and the exposure used. In this case, as a laboratory equipment for which a SWIR camera should be used, you used a converted camera that does not have the proper laboratory certification.

Shooting conditions have been added to the second chapter. The use of multispectral cameras would be a good help for preliminary research, but high-quality cameras are not yet available for museums in Russia.

Elemental analysis was performed using a TRACER 5 energy dispersive X-ray fluorescence spectrometer (Bruker). Please indicate how you separated the signals associated with the paint layers and the ground of the painting you studied. Is such separation of signals possible or impossible when using this type of spectrometer?

            Since the painting has the underlying layers of unknown thickness, the results should be verified using additional methods. The current study used no methods to estimate the depth of the elements based on the absorption intensity of the K, L, and M series, because the studied spot of the device is 1 cm in diameter. To determine the thicknesses on the database of the XRF spectrometer, it is necessary to carry out mapping with a spot of about 1 mm and a scanning shift.

FT-IR spectroscopy - please indicate the mass of the samples of the substance used for this type of analysis.

It is corrected

Optical microscopy - in addition to a simple enumeration of the instruments used, it is necessary to indicate at least the degree of magnification used in the analysis of the surface of the picture and sampling from the paint layer.

It is corrected

Radiography - this research method is not described at all, even the brand of the device you used is not indicated.

It is corrected.

“An X-ray integral image of the structure was obtained. An X-ray tube (Introvolt 100 VE, Promavtomatika) was placed under the object and directed to its front side. X-ray tube specifications are voltage 35 kV and current 4 mA. Consequently, the X-ray film (Agfa 100 NIF, Agfa) was over the object and directed to its reverse side”

Section 3.2. Observations on UV luminescence, IR radiation, radiography

Your reasoning in the context of evaluating the results of photography “about luminescent pigments, which in most cases are organic compounds” is methodically naive and incorrect - photography in the light of visible luminescence is not a method of chemical analysis and does not allow you to get the declared result.

It is corrected.

 Photography in the UV-induced visible luminescence (Fig. 3 (a)) revealed the heterogeneous structure of the blue sky; moreover, it allowed observing luminescent pigments which can be organic as well as some inorganic pigments (for example, cadmium red). This assumption was later confirmed by the results of X-ray elemental analysis and Fourier transform Infrared spectroscopy [24–26].

The authors' reasoning about the portrait that exists under the colorful layers of the studied painting is based on the results of X-ray analysis, which is not included in the description of the research methods used.

It is corrected

Section 3.3. Elemental analysis of imaging layers (XRF)

The results of the determination of pigments by the XRF method declared by you are incorrect and do not correspond to the research method used. Pigment determinations can be obtained by XRD or Raman spectroscopy, which can take the form of a non-destructive method. Modern XRF analysis with the determination of the quantitative elemental composition and elemental mapping allows us to assume the presence of certain types of pigments. But the authors defined only chemical elements, and only in the form of the simplest qualitative, not quantitative analysis. Operating with the results of the XRF method, in its simplest manifestation, you are actually engaged in the intuitive reconstruction of pigments, replacing the result of instrumental studies with your ideas about pigments. For example, the formula of lead white given by you can be confirmed only by the results of phase analysis carried out by the XRD method, or by the results of Raman spectroscopy, but not by the results of XRF analysis in its simplest qualitative form.

The wording regarding XRF results has been corrected. Chemical formulas are given as assumptions based on the main elements. Quantitative analysis based on the method of fundamental parameters or on the basis of built-in calibrations of inorganic compounds (supplied with the device) has many shortcomings and conventions. The purpose of our experiment was a preliminary study of the elemental composition of paint layers by XRF and a more detailed study using FTIR. Unfortunately, these are all devices for determining the chemical composition in the museum.

The results of the study of the paint layer by optical microscopy are not confirmed by anything except for verbal declarative statements. Two shots of medium quality green pigment (Figure 8) are clearly not enough.

XRF and FTIR spectra were added. A photo of the green pigment - volkonskoite for visual comparison is added to the fragment with polarizing microscopy.

Section 4. Conclusions.

Some of the conclusions stated in the paper do not correspond to the results of the study.

The authors claim that they have also identified pigments typical of 20th-century painting: lead-zinc white, red cadmium sulfide selenide, cadmium sulfide yellow, ultramarine blue, lithol red PR3, Hansa yellow PU3. But you could not get such a result, given that you used XRF in its simplest form as the main method for identifying pigments.

Added FTIR spectra.

In general, the work is more of an art history, rather than an interdisciplinary study. In the work, it is the aspect of art history that is most vivid and informative, and the instrumental research methods used by the authors are simple and uninformative, the methods of these studies are described in the work extremely briefly or not described at all (X-ray). Thus, the conclusions made by the authors on the composition of the pigments used by the master to create the work under study do not correspond to the results obtained and need to be corrected and improved.

            We tried to focus on the revealed portrait and connections with other basic studies of the artist's paintings. Therefore, we changed the title of the article to “Museum study of the painting “Hiroshima I” (1958) by Werner Tübke (1929-2004) in situ” instead of a multidisciplinary study. Pigment data has also been added.

Reviewer 2 Report

Manuscript ID: heritage-2410352

Title: Multidisciplinary study of the painting “Hiroshima I” (1958) by Werner Tübke (1929-2004)

1. Recommendation:

Reconsider after major revisions

2. Comments to the authors:

2.1 Overview and general recommendation

The paper " Multidisciplinary study of the painting “Hiroshima I” (1958) by Werner Tübke (1929-2004)" presents the analyses of a painting by Werner Tübek, an artist seldom investigated in the sources. The article is well-written and the information presented is well-organized. This work certainly contributes to increasing the knowledge of the technique of this artist; however, there are still some issues that need to be addressed to improve the quality.  For this reason, I recommend reconsidering it after major revisions. My concerns are explained in detail below.    

2.2 Major comments

This study is interesting and contributes to the research on modern painting. In my opinion, the work could benefit from a deeper discussion of the results obtained compared to the historical records (artist journal) and the few other studies on the paintings from the same author. Moreover, some results required a more detailed interpretation.

2.3 Minor comments

-       Page 1, line 40.  There are missing quotation marks.

-       Page 3, line 107. It could be better to use the term “UV-induced visible luminescence” The same in Page 4, line 134.

-       Page 4/5, Figure 2. Please add a scale to the images to have an idea of the actual size of the details. Also, please add the magnification in the caption.

-       Page 5, Figure 3b. It is not clear traits from the underlying painting, it could be helpful to increase the contrast or mark the details to make it easier to identify them. Also, in the same figure, it could be helpful to add the visible light photograph here for comparison.

-       Page 5, Figure 4a. In the caption, it could be better to use the term radiography instead of X-ray. Also, in the caption please add further information regarding the drawing reported in b, (collection, date, etc.).

-       Page 6. Line 194. Please change X-ray diffraction for X-ray fluorescence.

-       Page 6. Line 199. CaCO3, the number three should go in subscript.

-       Page 6. Line 199-200. Calcium sulfate dihydrate or just gypsum could be a better term.

-       Page 6. Figure 5. The term mapping could be misleading and confused with XRF mapping, maybe "scheme" could be better.

-       Page 7. Line 234. Please substitute infrared-Fourier spectroscopy for Fourier transform Infrared spectroscopy.

-       Page 8. Figure 7. You report Zn salts in the spectrum. This band can correspond to the presence of Zn carboxylates, probably in the amorphous form. It is well known that these species are very reactive and aggressive when present in the paint layers, producing several conservation problems. Please elaborate more on this and correlate it with the state of conservation of the painting.

-       Page 8. Lines 255-256. The phrase “It was opened at….” is not clear, please revise. Moreover, please elaborate more on the identification of this painting based on polarized microscopy. Do you have a database of minerals that you used for this characterization? If so, please add any reference images.

-       A table summarizing all the results obtained with the different techniques could be very useful in the conclusions. This can help to correlate all the data obtain according to the color area, etc.

-       Also, in the conclusions you mentioned the technical investigation of another painting. It could be very interesting for this work to make a Discussion section where a comparison between the results of this investigation with those results and what is mentioned in the meticulous journal by the artists. Does he mention the pigments used? The correlation between the names of the time and the pigments could be another contribution to the community studying modern paintings. 

English is ok. Some minor corrections are required.

Author Response

Manuscript ID: 2410352    
Title: Multidisciplinary study of the painting “Hiroshima I” (1958) by Werner Tübke (1929-2004)

First and foremost, we would like to thank the reviewers for their highly professional feedback which helped us improve the quality of the manuscript. Now, we would like to address the reviewers’ comments in the order of their appearance.

 --------Reviewer Comments--------

Reviewer 2:

Major comments

This study is interesting and contributes to the research on modern painting. In my opinion, the work could benefit from a deeper discussion of the results obtained compared to the historical records (artist journal) and the few other studies on the paintings from the same author. Moreover, some results required a more detailed interpretation.

A technological study of Tübke's painting is being carried out for the first time, with one exception: a similar study of the Peasant War panorama in Bad Frankenhausen was made in the 1990s. We are going to compare these two examinations and trace the evolution of Tübke's painting technology on this basis in the next article. We have described these plans in the discussion part of the current article.

Minor comments

Page 1, line 40.  There are missing quotation marks.

It is corrected.

Page 3, line 107. It could be better to use the term “UV-induced visible luminescence” The same in Page 4, line 134.

It is corrected.

Page 4/5, Figure 2. Please add a scale to the images to have an idea of the actual size of the details. Also, please add the magnification in the caption.

It is corrected.

Page 5, Figure 3b. It is not clear traits from the underlying painting, it could be helpful to increase the contrast or mark the details to make it easier to identify them. Also, in the same figure, it could be helpful to add the visible light photograph here for comparison.

            Photos with markers added (Fig. 4).

Page 5, Figure 4a. In the caption, it could be better to use the term radiography instead of X-ray. Also, in the caption please add further information regarding the drawing reported in b, (collection, date, etc.).

It is corrected.

Page 6. Line 194. Please change X-ray diffraction for X-ray fluorescence.

It is corrected.

Page 6. Line 199. CaCO3, the number three should go in subscript.

It is corrected.

Page 6. Line 199-200. Calcium sulfate dihydrate or just gypsum could be a better term.

It is corrected.

Page 6. Figure 5. The term mapping could be misleading and confused with XRF mapping, maybe "scheme" could be better.

It is corrected.

Page 7. Line 234. Please substitute infrared-Fourier spectroscopy for Fourier transform Infrared spectroscopy.

Сorrected throughout the text

Page 8. Figure 7. You report Zn salts in the spectrum. This band can correspond to the presence of Zn carboxylates, probably in the amorphous form. It is well known that these species are very reactive and aggressive when present in the paint layers, producing several conservation problems. Please elaborate more on this and correlate it with the state of conservation of the painting.

            Added some thoughts on this. Fortunately, the classical aggregates of Zn or Pb carboxylates are not visible by visual inspection. Detailed studies of this phenomenon are going well using microscopic techniques - mFTIR, mRaman, SEM ... In this article, we decided to limit ourselves to mentioning the problem for the time being, because, the main focus was on the field of art history.

Page 8. Lines 255-256. The phrase “It was opened at….” is not clear, please revise. Moreover, please elaborate more on the identification of this painting based on polarized microscopy. Do you have a database of minerals that you used for this characterization? If so, please add any reference images.

It is corrected. A photo of the green pigment - volkonskoite for visual comparison is added to the fragment with polarizing microscopy

A table summarizing all the results obtained with the different techniques could be very useful in the conclusions. This can help to correlate all the data obtain according to the color area, etc.

General information about pigments is presented in Table 2 before the Discussion section.

Also, in the conclusions you mentioned the technical investigation of another painting. It could be very interesting for this work to make a Discussion section where a comparison between the results of this investigation with those results and what is mentioned in the meticulous journal by the artists. Does he mention the pigments used? The correlation between the names of the time and the pigments could be another contribution to the community studying modern paintings.       

We have added a discussion section to the article, in which we mention a technological study of the panorama painting in Germany in the 1990s. In the future we plan to carry out a more precise analysis of the pigments and to compare the two examinations carried out at the Russian Museum and at the Panorama Museum in Bad Frankenhausen. The data will certainly be considered in the context of the artist's diary entries. This should help us understand the evolution of the artist's painting and his connection to his predecessors at the level of technology. The volume of material to be studied is too large and goes beyond the objectives of this article (the study of the painting Hiroshima I).

Reviewer 3 Report

Summary

In the article “Multidisciplinary study of the painting “Hiroshima I” (1958) by Werner Tübke (1929-2004)”, a case study is presented in which a multi-analytical methodology is applied to identify the pigments in a relevant contemporary painting by Werner Tübke. The methodology employed is not novel (combination of Vis and UV photography, IR reflectography, radiography, pXRF and FT-IR) but the study is nonetheless interesting due to the limited material-technical information existing on the work of this German painter. The analysis allowed to identify a combination of traditional and modern synthetic pigments. This information will be potentially relevant to further study the evolution of materials and techniques in the art of Tübke and his contemporaries. In addition, the radiographic analysis also allowed to discover an antecedent portrait underlying the actual painting.

General remarks

Generally speaking, the results obtained in this study are novel and relevant for the heritage science, conservation/restoration and art history communities. However, the discussion of the results should be improved both in terms of content and form before this manuscript can be accepted for publication.

You can find some more specific remarks below:

1.       In the manuscript, conclusions are drawn on the basis of a number of scientific analyses. However, only a very small part of the data is actually shown to the readers. The "results" section of the manuscript should be significantly improved. For example, red and yellow synthetic organic pigments (PR3 and PY3 respectively) are identified based on FTIR analysis. However, no spectra is shown that actually supports this claim. The same goes for the identification of Ultramarine. According to the text, this is also based on FTIR data, but the spectra are not shown. It is very important to add in the manuscript all the data that you actually discuss in the text, and especially the ones you use to confirm or confute a certain hypothesis. Some of the data could be added to Supplementary information, but should be nonetheless made available to the readers.

2.       The structure of the manuscript should be improved. The actual interpretation of the results is discussed in large part only in the conclusions. This translates to a short “results” section and a very long “Conclusions” section. I suggest moving the discussion of the results to the “results” section and to keep the “conclusions” as focused as possible, to make the message clearer.

3.       In the abstract (line 18), it is mentioned that “an integrated approach was used using non–invasive control methods”. However, both ATR-FTIR and polarization microscopy were performed on samples (i.e. the methodology was invasive). The “non-invasive” claim should be removed.

4.       The painting Hiroshima I by Tubke is part of a series of 3 paintings on the same theme (Hiroshima II and Hiroshima III). This should be mentioned/discussed in the text.

5.       The green pigment is described as volkonskoite. This is quite interesting since, according to the authors, the artist used this unusual pigment in other of his works. It is not clear, however, which results support this hypothesis. In the text, it seems like the final identification is based on polarization microscopy. Could you please explain how the polarization microscopy results allow you to identify this pigment as volkonskoite rather than Cr oxide? Do the FTIR results support your interpretation? Since volkonskoite is an aluminosilicate, there should be some clear markers in the IR spectra.

6.       As clearly shown by the radiographic analysis, a second painting is present underneath Hiroshima I. The composition of the underlying painting would therefore certainly affect the results of your XRF analysis. Did you take into account the possible contribution of the underlying paint layers in your XRF spectra? Since you had the opportunity to extract some samples from the painting, did you prepare any cross-sections to study the stratigraphy of the painting (e.g. with optical microscopy and SEM-EDX)? SEM-EDX, in particular, would be a better choice to investigate the elemental composition of such a “multi-layered” painting.

7.       The FTIR spectrum in figure 7 shows some contribution of metal soaps. This is not discussed in the text. In such a recent painting metal soaps could be both degradation products but also plasticizers or other additives. Do you see any contribution of metal sopas in other IR spectra? Do you think they are degradation products due to the interaction of ZnO or lead white and the binding medium? As this could be relevant for conservation purposes, it should be mentioned in the text.

8.       In abstract, introduction and conclusions it is mentioned that this study investigated both pigments and binding medium/media. However, only the results regarding the pigments are described in the text. Can you please comment on the composition of the binding medium in the different areas of the painting? Is it actually a siccative oil as mentioned by the artist? Is it always the same in all the areas? This should be discussed in the text.

9.       The correct full name of FTIR is Fourier-transform infrared spectroscopy. If a different name is used in the text, this should be corrected (e.g. in the abstract it is called infrared-Fourier spectroscopy).

10.   You seem quite certain that the underlying portrait is of Tubke’s father. Do you have any more information to support this claim? (E.g. was his father a common subject in his paintings? Was he known to paint portraits of strangers as well, or only of his close family? Etc.). There certainly is some similarity between the portrait and the drawing of Tubke’s father you show in the text, but any additional info would strengthen your claim.

11.   The language used is sometimes incorrect or simply awkward, I suggest revising the whole manuscript and trying to keep the language concise and simple. Few examples:

-          Line 132: “Photofixation of images in visible light” should be changed to  “Pictures in visible light”

-          Line 180: it is not clear what you mean by “overlooked”. Reformulate

-          Line 196: “this canvas was made of lead white..” should be changed to e.g. “ was coated with a ground/preparation layer containing lead white” or simply “ the ground layer contains lead white”

-          Line 199: “two-water gypsum” should be changed to just gypsum (Gypsum is calcium sulfate dihydrate, so no need to mention the “two-water” part).

-          Line 256: “it was opened” do you maybe mean “it was introduced”?

-          Lines 92-103: very confusing paragraph. The message would be clearer if you tried to keep it more concise.

-          Etc.

Point corrections

·       Line 194: X-ray fluorescence, not diffraction

·       Line 204: “CdX” should be changed to “CdS”

·       Line 248: the “absorption bands of Zinc” should be changed to “Zinc white” or “Zinc oxide”

The language of the manuscript should be revised. The wording is sometimes awkward or incorrect. Moreover, long and complex paragraphs are used to describe rather simple concepts, especially in introduction and conclusions. I suggest revising the whole manuscript and trying to keep the language as concise and simple as possible. Some more specific suggestions are given in the section “Comments and suggestions for authors”.

Author Response

Manuscript ID: 2410352    
Title: Multidisciplinary study of the painting “Hiroshima I” (1958) by Werner Tübke (1929-2004)

First and foremost, we would like to thank the reviewers for their highly professional feedback which helped us improve the quality of the manuscript. Now, we would like to address the reviewers’ comments in the order of their appearance.

 --------Reviewer Comments--------

Reviewer 3:

In the manuscript, conclusions are drawn on the basis of a number of scientific analyses. However, only a very small part of the data is actually shown to the readers. The "results" section of the manuscript should be significantly improved. For example, red and yellow synthetic organic pigments (PR3 and PY3 respectively) are identified based on FTIR analysis. However, no spectra is shown that actually supports this claim. The same goes for the identification of Ultramarine. According to the text, this is also based on FTIR data, but the spectra are not shown. It is very important to add in the manuscript all the data that you actually discuss in the text, and especially the ones you use to confirm or confute a certain hypothesis. Some of the data could be added to Supplementary information, but should be nonetheless made available to the readers.

XRF and FTIR spectra were added. A photo of the green pigment - volkonskoite for visual comparison is added to the fragment with polarizing microscopy.

The structure of the manuscript should be improved. The actual interpretation of the results is discussed in large part only in the conclusions. This translates to a short “results” section and a very long “Conclusions” section. I suggest moving the discussion of the results to the “results” section and to keep the “conclusions” as focused as possible, to make the message clearer.

            Divided into two sections – 4. Discussion / 5. Conclusions

In the abstract (line 18), it is mentioned that “an integrated approach was used using non–invasive control methods”. However, both ATR-FTIR and polarization microscopy were performed on samples (i.e. the methodology was invasive). The “non-invasive” claim should be removed.

It is corrected.

The painting Hiroshima I by Tubke is part of a series of 3 paintings on the same theme (Hiroshima II and Hiroshima III). This should be mentioned/discussed in the text.

It is corrected.

The green pigment is described as volkonskoite. This is quite interesting since, according to the authors, the artist used this unusual pigment in other of his works. It is not clear, however, which results support this hypothesis. In the text, it seems like the final identification is based on polarization microscopy. Could you please explain how the polarization microscopy results allow you to identify this pigment as volkonskoite rather than Cr oxide? Do the FTIR results support your interpretation? Since volkonskoite is an aluminosilicate, there should be some clear markers in the IR spectra.

            Unfortunately, based on the research results, we cannot give a definite answer: we see Cr in the XRF spectrum; the FTIR spectrum shows a slight broadening near the barium sulfate vibrations in the absorption region of silicate groups, as well as the absence of pronounced Cr-O stretching vibrations. It is possible that the pigment concentration is too low to be identified by FTIR.

Because there are no other methods for analysis, then only polarizing microscopy remains. We have added a photo from our reference database for comparison, but this does not allow us to accurately judge the chemical composition of the pigment. We will try to clarify Volkonskoit using other methods.

As clearly shown by the radiographic analysis, a second painting is present underneath Hiroshima I. The composition of the underlying painting would therefore certainly affect the results of your XRF analysis. Did you take into account the possible contribution of the underlying paint layers in your XRF spectra? Since you had the opportunity to extract some samples from the painting, did you prepare any cross-sections to study the stratigraphy of the painting (e.g. with optical microscopy and SEM-EDX)? SEM-EDX, in particular, would be a better choice to investigate the elemental composition of such a “multi-layered” painting.

            Because the underlying layer was detected, we did not use different options for quantitative analysis of elements based on XRF. And also to clarify all pigments, FTIR was used to analyze microprobes. Unfortunately, we were not allowed to get a sample with paint layers down to the primer to prepare the section. Therefore, the main emphasis is on art history. We also changed the title of the article, because. this is a museum study, and we do not have SEM, XRD, Multispectral camera in stock.

The FTIR spectrum in figure 7 shows some contribution of metal soaps. This is not discussed in the text. In such a recent painting metal soaps could be both degradation products but also plasticizers or other additives. Do you see any contribution of metal sopas in other IR spectra? Do you think they are degradation products due to the interaction of ZnO or lead white and the binding medium? As this could be relevant for conservation purposes, it should be mentioned in the text.

Added some thoughts on this in the text. Fortunately, the classical aggregates of Zn or Pb carboxylates are not visible by visual inspection. Detailed studies of this phenomenon are going well using microscopic techniques - mFTIR, mRaman, SEM ... In this article, we decided to limit ourselves to mentioning the problem for the time being, because, the main focus was on the field of art history.

In abstract, introduction and conclusions it is mentioned that this study investigated both pigments and binding medium/media. However, only the results regarding the pigments are described in the text. Can you please comment on the composition of the binding medium in the different areas of the painting? Is it actually a siccative oil as mentioned by the artist? Is it always the same in all the areas? This should be discussed in the text.

An oil binder is present on all spectra of paint layers. We have added information about the presence of an oil binder and described the main fluctuations in the FTIR spectrum. There are no absorption peaks of unsaturated carboxylic acids in the spectrum, which makes it possible to judge the change in the chemical composition of the binder. We could judge the type of oil based on the data of the mass spectra after chromatography.

The correct full name of FTIR is Fourier-transform infrared spectroscopy. If a different name is used in the text, this should be corrected (e.g. in the abstract it is called infrared-Fourier spectroscopy).

It is corrected.

You seem quite certain that the underlying portrait is of Tubke’s father. Do you have any more information to support this claim? (E.g. was his father a common subject in his paintings? Was he known to paint portraits of strangers as well, or only of his close family? Etc.). There certainly is some similarity between the portrait and the drawing of Tubke’s father you show in the text, but any additional info would strengthen your claim.

We have added another graphic work by Werner Tubke depicting his father (Fig. 5). Only these two works depicting his father are known at this time.

The language used is sometimes incorrect or simply awkward, I suggest revising the whole manuscript and trying to keep the language concise and simple. Few examples:

Line 132: “Photofixation of images in visible light” should be changed to  “Pictures in visible light”

It is corrected.

Line 180: it is not clear what you mean by “overlooked”. Reformulate

Line 196: “this canvas was made of lead white..” should be changed to e.g. “ was coated with a ground/preparation layer containing lead white” or simply “ the ground layer contains lead white”

It is corrected.

Line 199: “two-water gypsum” should be changed to just gypsum (Gypsum is calcium sulfate dihydrate, so no need to mention the “two-water” part).

It is corrected.

Line 256: “it was opened” do you maybe mean “it was introduced”?

It is corrected.

Lines 92-103: very confusing paragraph. The message would be clearer if you tried to keep it more concise.

-          Etc.

Point corrections

Line 194: X-ray fluorescence, not diffraction

It is corrected.

Line 204: “CdX” should be changed to “CdS”

It is corrected.

Line 248: the “absorption bands of Zinc” should be changed to “Zinc white” or “Zinc oxide”

It is corrected.

The language of the manuscript should be revised. The wording is sometimes awkward or incorrect. Moreover, long and complex paragraphs are used to describe rather simple concepts, especially in introduction and conclusions. I suggest revising the whole manuscript and trying to keep the language as concise and simple as possible. Some more specific suggestions are given in the section “Comments and suggestions for authors”.

Worked with a professional translator to improve the text.

Reviewer 4 Report

-The authors proposed that the paper represents an integrated technical study without conducting the important investigated and analytical techniques as cross section, GC- MS, Raman for confirming the pigments identification. 

- The authors referred some information in the highlight part about the identification  of materials used in tubke painting  with no results mentioned as the identification of the binding medium, painting technique "layers" and conservation state as example. 

- The abstract does not guide the reader about results of the paper with misleading the reader about using non-invasive methods while using FTIR with crushed sample and using polarizing microscopy should deteriorate the sample. 

- The first paragraph of the introduction does not refer to the paper content at all with using pronouns?!.... the pronoun "I" relates to whom?

- on  page 3, line 104 what do you mean by Tubke technology?! ... artistic features of the painter.

- A review of English should be done by native speaker as the authors sometimes use German and Italian terms inside the text involving them with English as "the alla prima " for example with using some terms that are not related to the oil painting as whitewash instead of priming layer also technical study is preferred than technological study. 

- Radiography imaging is not obvious as it revealed the underlying part without indicated the whole image characterizing the place of man face.

on page 6 line 196... This canvas was made of lead white! ... Proofreading is essential and line 199 " or two water gypsum?!

- XRF does not confirm the pigments identification so it should be mentioned that the existence of element 1, element 2 and etc suggests the the pigment could be ..... as some pigment elements referred to the possibility of pigment mixtures. 

- Conclusion includes new information so the authors need to discuss the results in a discussion part and conclude the paper content with the future recommendation in the conclusion part.

The full paper needs extensive proofreading and English revision by a native speaker

Author Response

Manuscript ID: 2410352    
Title: Multidisciplinary study of the painting “Hiroshima I” (1958) by Werner Tübke (1929-2004)

First and foremost, we would like to thank the reviewers for their highly professional feedback which helped us improve the quality of the manuscript. Now, we would like to address the reviewers’ comments in the order of their appearance.

 --------Reviewer Comments--------

Reviewer 4:

The authors proposed that the paper represents an integrated technical study without conducting the important investigated and analytical techniques as cross section, GC- MS, Raman for confirming the pigments identification.

            We have changed the title of the article to “ Museum study of the painting “Hiroshima I” (1958) by Werner Tübke (1929-2004) in situ”

The authors referred some information in the highlight part about the identification  of materials used in tubke painting  with no results mentioned as the identification of the binding medium, painting technique "layers" and conservation state as example.

Added information and spectra regarding the identification of pigments, and also expanded the data along the text.

The abstract does not guide the reader about results of the paper with misleading the reader about using non-invasive methods while using FTIR with crushed sample and using polarizing microscopy should deteriorate the sample.

            Abstract has been rewritten

The first paragraph of the introduction does not refer to the paper content at all with using pronouns?!.... the pronoun "I" relates to whom?

The structure of the article has been revised and supplemented

on  page 3, line 104 what do you mean by Tubke technology?! ... artistic features of the painter.

It is corrected.

A review of English should be done by native speaker as the authors sometimes use German and Italian terms inside the text involving them with English as "the alla prima " for example with using some terms that are not related to the oil painting as whitewash instead of priming layer also technical study is preferred than technological study.

Worked with a professional translator to improve the text.

Radiography imaging is not obvious as it revealed the underlying part without indicated the whole image characterizing the place of man face.

Photos with markers added (Fig. 4).

on page 6 line 196... This canvas was made of lead white! ... Proofreading is essential and line 199 " or two water gypsum?!

It is corrected.

XRF does not confirm the pigments identification so it should be mentioned that the existence of element 1, element 2 and etc suggests the the pigment could be ..... as some pigment elements referred to the possibility of pigment mixtures.

The wording regarding XRF results has been corrected. Chemical formulas are given as assumptions based on the main elements. Quantitative analysis based on the method of fundamental parameters or on the basis of built-in calibrations of inorganic compounds (supplied with the device) has many shortcomings and conventions. The purpose of our experiment was a preliminary study of the elemental composition of paint layers by XRF and a more detailed study using FTIR. Unfortunately, these are all devices for determining the chemical composition in the museum.

Conclusion includes new information so the authors need to discuss the results in a discussion part and conclude the paper content with the future recommendation in the conclusion part.

Divided into two sections – 4. Discussion / 5. Conclusions

Comments on the Quality of English Language

The full paper needs extensive proofreading and English revision by a native speaker

Worked with a professional translator to improve the text.

Round 2

Reviewer 1 Report

Dear Colleagues! Thank you for your work. Many of your conclusions on the interpretation of pigments are vague and ambiguous, but they fully reflect the advantages and disadvantages of the methodological approach you have chosen. It is the debatability of the results of your instrumental studies, in my opinion, that is interesting, since it illustrates the research capabilities of the equipment you used.

Author Response

Dear Colleagues! Thank you for your work. Many of your conclusions on the interpretation of pigments are vague and ambiguous, but they fully reflect the advantages and disadvantages of the methodological approach you have chosen. It is the debatability of the results of your instrumental studies, in my opinion, that is interesting, since it illustrates the research capabilities of the equipment you used.

Response: Thank you! In the next study we will try to take your comment into account, improve the methods, and come up with new, more accurate results.

Reviewer 2 Report

I want to thank the authors for considering the comments of the previous round. 

The paper has been considerably improved and all the comments have been addressed. There are still a few issues that could be modified:

Title: The authors have modified the paper's title; however, "Museum study" seems not to be the best choice, please consider changing it to: "In-situ study of the painting...."

Page 1, Line 41: Please add German Democratic Republic (GDR) to be more precise.

Page 5. Lines 199-200. The phrase "Aggregates of lead and zinc salts of carboxylic acids..." can be moved to the section where the evidence of metal carboxylates is discussed, ie. Page 12 lines 346-350.

Page 12. Lines 346-350. It is difficult to be sure since it is impossible to see the exact wavenumber of the bands in your spectra but it seems there is more than one type of metal carboxylate in the painting, namely, amorphous in (probably Pb in the white areas) 10a blue line, and crystalline? in 10b (band at around 1538 cm-1? Zn?). Consider making some comments on this.

Author Response

First and foremost, we would like to thank the reviewers for their highly professional feedback which helped us improve the quality of the manuscript. Now, we would like to address the reviewers’ comments in the order of their appearance.

Title: The authors have modified the paper's title; however, "Museum study" seems not to be the best choice, please consider changing it to: "In-situ study of the painting...."

Response: Corrected the title of the article to “In-situ study of the painting “Hiroshima I” (1958) by Werner Tübke (1929-2004)”

Page 1, Line 41: Please add German Democratic Republic (GDR) to be more precise.

Response: It is corrected.

Page 5. Lines 199-200. The phrase "Aggregates of lead and zinc salts of carboxylic acids..." can be moved to the section where the evidence of metal carboxylates is discussed, ie. Page 12 lines 346-350.

Response: corrected 

Page 12. Lines 346-350. It is difficult to be sure since it is impossible to see the exact wavenumber of the bands in your spectra but it seems there is more than one type of metal carboxylate in the painting, namely, amorphous in (probably Pb in the white areas) 10a blue line, and crystalline? in 10b (band at around 1538 cm-1? Zn?). Consider making some comments on this.

Response: Added a few suggestions about carboxylates

Reviewer 3 Report

The authors thoroughly addressed the previous comments and significantly improved the content and form of the manuscript. I believe the manuscript can be published in the present form, I just have one final comment/suggestion left for the authors.

In this new revised version of the manuscript, Hg has been added in the list of elements identified in red areas (see Table 1). Does that mean that the painter not only used Cadmium red and organic reds, but also cinnabar? This is not discussed anywhere in the text. In the results section, you actually mention that Cadmium red was introduced to replace HgS (historically), so I believe it would be interesting if you found these two pigments mixed in your painting. I really think you should add a very brief comment on that in the tmanuscript, or it might result confusing for the readers to just find Hg mentioned in the Table and nowhere else in the text.

Author Response

First and foremost, we would like to thank the reviewers for their highly professional feedback which helped us improve the quality of the manuscript. Now, we would like to address the reviewers’ comments in the order of their appearance.

In this new revised version of the manuscript, Hg has been added in the list of elements identified in red areas (see Table 1). Does that mean that the painter not only used Cadmium red and organic reds, but also cinnabar? This is not discussed anywhere in the text. In the results section, you actually mention that Cadmium red was introduced to replace HgS (historically), so I believe it would be interesting if you found these two pigments mixed in your painting. I really think you should add a very brief comment on that in the tmanuscript, or it might result confusing for the readers to just find Hg mentioned in the Table and nowhere else in the text.

Response: corrected. Added a few suggestions about the presence of CdSe and HgS. 

Reviewer 4 Report

Highlights the oil binder type is not identified so please remove it ….  Linseed oil, walnut oil or

-           

Replace the word of Technological by technical

-          On line 49, The intermediate coat of varnish " damask?" do you mean dammar or mastic varnish?

-          On line 50 … in the alla prima technique? Alla prima is Italian word so please translate into English

-          On lines  190 and 350 please do not use pronouns as "us"  … it does not make sense in academic writing.

   On line 49, The intermediate coat of varnish " damask?" do you mean dammar or mastic varnish?

-          On line 50 … in the alla prima technique? Alla prima is Italian word so please translate into English

-          On lines  190 and 350 please do not use pronouns as "us"  … it does not make sense in academic writing.

Author Response

First and foremost, we would like to thank the reviewers for their highly professional feedback which helped us improve the quality of the manuscript. Now, we would like to address the reviewers’ comments in the order of their appearance.

Highlights the oil binder type is not identified so please remove it ….  Linseed oil, walnut oil or

Response: we identified oil binder based on FTIR data and didn’t write it’s type across the text. 

- Replace the word of Technological by technical

Response: corrected

-          On line 49, The intermediate coat of varnish " damask?" do you mean dammar or mastic varnish?

It is corrected.

-          On line 50 … in the alla prima technique? Alla prima is Italian word so please translate into English

Response: it is corrected.

-          On lines  190 and 350 please do not use pronouns as "us"  … it does not make sense in academic writing.

Response: corrected

Comments on the Quality of English Language 

   On line 49, The intermediate coat of varnish " damask?" do you mean dammar or mastic varnish?

It is corrected, Tübke meant dammar.

-          On line 50 … in the alla prima technique? Alla prima is Italian word so please translate into English

It is corrected.

-          On lines  190 and 350 please do not use pronouns as "us"  … it does not make sense in academic writing.

It is corrected.